# Relations between Stress and Depressive Symptoms in Psychiatric Nurses: The Mediating Effects of Sleep Quality and Occupational Burnout

**DOI:** 10.3390/ijerph18147327

**Published:** 2021-07-08

**Authors:** Hsiu-Fen Hsieh, Yi Liu, Hsin-Tien Hsu, Shu-Ching Ma, Hsiu-Hung Wang, Chih-Hung Ko

**Affiliations:** 1School of Nursing, College of Nursing, Kaohsiung Medical University, No.100, Shih-Chuan 1st Road, Kaohsiung 807, Taiwan; hsiufen96@gmail.com (H.-F.H.); gn94yliu@kmu.edu.tw (Y.L.); hthsu@kmu.edu.tw (H.-T.H.); 2Department of Medical Research, Kaohsiung Medical University Hospital, Kaohsiung Medical University, Kaohsiung 807, Taiwan; 3Nursing Department, Chi-Mei Medical Center, Tainan 71004, Taiwan; 300004@mail.chimei.org.tw; 4College of Humanities and Social Science, Southern Taiwan University of Science and Technology, Tainan 71005, Taiwan; 5Department of Psychiatry, Kaohsiung Medical University Hospital, College of Medicine, Kaohsiung Medical University, Kaohsiung 807, Taiwan; chihhungko@gmail.com

**Keywords:** stress, occupational burnout, quality of sleep, depressive symptoms, psychiatric nurses

## Abstract

This study examines the parallel multiple mediators of quality of sleep and occupational burnout between perceived stress and depressive symptoms in psychiatric nurses. Nurses are more likely to experience depression, anxiety, decreased job satisfaction, and reduced organizational loyalty as a result of the stressful work environment and heavy workload. A total of 248 psychiatric ward (PW) nurses participated in this cross-sectional survey study. Structural equation modelling was used for data analysis. In the model of parallel multiple mediators for depressive symptoms, quality of sleep and occupational burnout played mediating roles, and these two mediators strengthened the effect of stress on depressive symptoms, with the final model showing a good fit. Stress, occupational burnout, and quality of sleep explained 46.0% of the variance in psychiatric nurses’ depressive symptoms. Stress had no significantly direct effect on psychiatric nurses’ depressive symptoms, but it had a completed mediation effect on their depressive symptoms through occupational burnout and quality of sleep. This study showed that reduction of occupational burnout and improvement of quality of sleep play important roles against depressive symptoms among PW nurses. Healthcare managers should provide PW nurses with a better environment for improving quality of sleep and reducing occupational burnout.

## 1. Introduction

Healthcare providers at hospitals are likely to experience depression, anxiety, decreased job satisfaction, and reduced organizational loyalty as a result of the stressful work environment [1]. As frontline healthcare providers, nurses’ heavy workload can make them become overworked and stressed, so the nature of nursing care consequently exposes them to higher risk of suffering from depression, anxiety, and stress [2]. This higher risk of developing depressive symptoms has prevalence rates ranging from 11.0–32.4% [1,2]. It has been documented that 30% of nurses have depressive symptoms, compared to only 4% in the general population [3]. For nurses in psychiatric wards (PW)s, the prevalence rates range from 52.7–74.9% [4,5].

A previous study has suggested that nursing work stress originates from multiple sources, including excessive workload, inadequate respect and recognition, conflicts with other colleagues, unfriendly supervisors, insufficient support systems, and the need to cope with emotional patients and their families [6]. Under such a demanding job, the ongoing occupational stress faced by nurses could have a great impact on their mental health [7], and this may worsen their subjective psychological well-being [8]. A study conducted in Taiwan reported a prevalence of sleep disturbance of 70.1% among nurses [9]. Sleep disturbance, which is the most common effect of shift work or long-term night shifts, compromises the quality of sleep of nurses due to biological clock disorders [10]. Previous research has shown that poor sleep is highly correlated with shift work disorder and can lead to severe depressive symptoms [11,12], while one study found that sleep quality had an indirect effect on depressive symptoms, indicating sleep quality is a mediator [13].

In addition to the abovementioned problems, the main job function of PW nurses is to provide care to patients with complex and challenging conditions who have been admitted to the hospital primarily because of severe mental illness and instability. Not only are these patients more demanding, but some also have judicial problems or a tendency toward violence and suicide. The unstable behavior of these patients due to mental disorder, along with a challenging and unpredictable work environment, places PW staff under significant stress. Among all the staff, PW nurses work most at the forefront of patient care and therefore have longer and more frequent contact with patients, making PW nurses more susceptible to professional stress, tension, dissatisfaction, and occupational burnout [14].

Occupational burnout, stress, quality of sleep, and depressive symptoms might form a complex relationship of mutual influence among PW nurses. A hypothesized model is presented in Figure 1. The model identifies the relationships among stress, occupational burnout, quality of sleep, and depressive symptoms. Model testing was carried out to examine those relationships.

This study aimed to (a) test the fit of the hypothesized model for PW nurses’ occupational burnout and quality of sleep; and (b) determine the extent to which stress, quality of sleep, and occupational burnout predicted depressive symptoms among PW nurses.

## 2. Methods

### 2.1. Design and Setting

This is a cross-sectional design study, and our participants were recruited using a convenience sampling from psychiatric wards (PWs) in two medical centers and three mental hospitals in Southern Taiwan.

### 2.2. Participants

The sample size was calculated by the method for covariance structure modeling [15], and the minimum sample size for our model structure was 212. A total of 260 PW nurses participated in our study, and 248 of them completed all questionnaires.

### 2.3. Data Collection and Ethical Considerations

After receiving approval from the Institutional Review Board (KMUH-IRB-20170175), the principal investigator (the first author), contacted the nursing department in each hospital, then the principal investigator went to the PWs in each hospital and explained the purpose of the study to all voluntary participants. The data were collected from July 2017 to June 2018. Each PW nurse received an envelope contained informed consent form and questionnaires. Once the questionnaires were completed, it could be sealed in the envelope by participants to be collected by the principal investigator. Each participant received a NT$ 200 (US$ 7.15) gift voucher in appreciation for their participation.

### 2.4. Instruments

In addition to demographic data, five latent variables (personal characteristics, stress, quality of sleep, occupational burnout, and depressive symptoms) were included in our hypothesized model.

#### 2.4.1. Chinese Pittsburgh Sleep Quality Index

The Pittsburgh Sleep Quality Index (PSQI) was developed for assessing quality of sleep [16]. The Chinese Pittsburgh Sleep Quality Index (CPSQI) was translated into Mandarin Chinese by a bilingual researcher from Taiwan [17]. This scale is composed of 19 items and seven subcategories. Each of these questions must be answered on a four-point scale ranging from “never” to “three times or more a week”, with each item in the CPSQI scale being scored between 0 and 3. The subjective rating of the participants’ sleep quality, sleep latency, sleep duration, sleep efficiency, sleep disturbance, use of sleeping medication, and daytime dysfunction (four-point scale from “very good” to “very bad”) and the sum of these items of the CPSQI scale range is scored from 0 to 21, with higher scores suggesting poorer sleep quality, and the respondents would be recognized as poor sleepers if they received a CPSQI score ≥5 [17].

#### 2.4.2. Stress

In this study, stress was evaluated using the Brief Stress Scale (BSS) developed for assessing stress in Taiwan [18]. The scale includes six items covering various work stress situations and job characteristics, such as work control, workload, and workplace relationships. Responses on the items are based on a five-point Likert scale (5 = “Always (5–6 days per week),” 4 = “Often (3–4 days per week),” 3 = “Occasionally (1–2 days per week),” 2 = “Rarely (1 day every few weeks),” and 1 = “Never”). The raw score is divided by three and multiplied by 10 to obtain the total score. A total score ≥ 60 indicates a high degree of stress, 40–59 indicates a moderate degree of stress, and <40 indicates a low degree of stress.

#### 2.4.3. The Center for Epidemiologic Studies Depression

The Center for Epidemiologic Studies Depression (CES-D) has been commonly used to evaluate the severity of depression with for a wide range of ages [19]. The Chinese version of CES-D was developed, with a Cronbach’s alpha of 0.86, and it was used as the screening measure of depressive symptoms with a Cronbach’s alpha of 0.86 in this study. This scale consists of 20 items and each item uses a four-point Likert scale to access levels of agreement to each statement. The total scores of the Chinese version of CES-D range from 0 to 60 and the cut-off point is greater than or equal to 15 (defined as the depressive group) [20].

#### 2.4.4. Occupational Burnout Inventory

We adopted the Chinese edition of the Occupational Burnout Inventory (OBI), which was translated from the Copenhagen Burnout Inventory [21] and the effort–reward imbalance model [22], containing the following subscales: personal burnout, work-related burnout, client-related burnout, and over-commitment to work. Each item is rated on a five-point Likert from 0 points (strongly disagree) to 4 points (strongly agree). Personal burnout ranges 0–100, work-related burnout range 0–100, client-related burnout range 0–100, and over-commitment to work range 0–100. The total score ranges from 0 to 400 points. The total score was positively correlated with the frequency of occupational burnout. The Cronbach’s alpha values for the four subscales were as follows: personal burnout (0.92), work-related burnout (0.91), over-commitment (0.84), and client-related burnout (0.90), and the Cronbach’s alpha of the overall scale was 0.94 [23].

### 2.5. Data Analysis

We used SPSS version 21.0 (IBM Corp., Armonk, NY, USA) and AMOS version 21.0 (SPSS Inc., Chicago, IL, USA) to analyze our data. Structural equation modeling was used to test model fit. In this study, the model fit was examined using chi-square (χ^2^), goodness of fit index (GFI), adjusted goodness of fit index (AGFI), and the root mean square error of approximation (RMSEA). A χ^2^ value without significance, GFI and AGFI greater than 0.90, and RMSEA less than 0.05 indicated a good model fit. The Sobel test is a method of testing the significance of a mediation effect, and a Z value ≧ 1.96 showed that meditation might exist [24].

## 3. Results

### 3.1. Participant Characteristics

Participants’ ages ranged from 20 to 52 years, with a mean = 32.98(SD ± 8.25), all participants being female 100% (*n* = 248), unmarried 61.7% (*n* = 153), graduated from college 94.4% (*n* = 233), and working in rotating shifts (74.1%). The average duration of work experience was 10.56 (SD ± 8.01) years. More than half of them had religious beliefs 60.5% (*n* = 150). In addition, the score of depressive symptoms was 14.49 (SD ± 9.27), the score of quality of sleep was 7.04 (SD ± 3.19) with a higher score indicating poorer quality of sleep (score ≥ 5); the score of stress was 152.28 (SD ± 86.38), and the score of occupational burnout was 179.13 (SD ± 79.07). The Cronbach’s alpha of the CPSQI, OBI, BSS, and CES-D were 0.91, 0.96, 0.91, and 0.91 respectively.

### 3.2. Preliminary Analysis

Correlational analysis was used to confirm significantly related factors leading to depressive symptoms. PW nurses’ stress was not significantly related to age (r = −0.053, *p* = 0.402), education (r = −0.043, *p* = 0.499), or seniority (r = −0.035, *p* =0.582). In addition, PW nurses’ depressive symptoms were not significantly related to age (r = −0.034, *p* = 0.596), education (r = −0.072, *p* = 0.258), seniority (r = −0.039, *p* = 0.544), or sleep efficiency (r =−0.089, *p* = 0.165). However, depressive symptoms were significantly related to sleep quality (r =0.376, *p* < 0.001), sleep latency (r =0.268, *p* < 0.001), sleep duration (r = 0.169, *p* < 0.001), sleep disturbance (r = 0.471, *p* < 0.001), use of sleeping medication (r = 0.159, *p* = 0.012), daytime dysfunction (r = 0.272, *p* < 0.001), and overall quality of sleep (r = 0.395, *p* < 0.001). In addition, depressive symptoms were significantly related to personal burnout (r = 0.591, *p* < 0.001), work-related burnout (r = 0.504, *p* < 0.001), client-related burnout (r = 0.394, *p* < 0.001), and over-commitment to work (r = 0.344, *p* < 0.001). Depressive symptoms were significantly related to stress (r = 0.456, *p* < 0.001). Item parcels may be preferred over individual items as indicators for a variety of reasons [25]. In our study, item parcels and modification indices [26] decided whether to delete the measured variable and goodness of fit for the model (Figure 2).

### 3.3. Structural Equation Model

A good fit for our model was obtained with χ^2^ = 124.989, degree of freedom = 81, *p* = 0.001, normed chi square = 1.543, GFI = 0.939, AGFI = 0.910, CFI = 0.973, NNFI = 0.965, IFI = 0.974, and RMSEA = 0.047. The structural relationships with standardized coefficients are presented in Figure 2. We used the Sobel test to examine indirect effects in a structural equation model (SEM). This showed a mediating effect of quality of sleep for the relationship between stress and depressive symptoms (z = 2.931, *p* = 0.001), and another mediating effect of occupational burnout for the relationship between stress and depressive symptoms (z = 3.285, *p* = 0.001), indicating that PW nurses’ quality of sleep and occupational burnout were significant determinants of depressive symptoms. The path from stress to depressive symptoms was significant (z = 1.755, *p* = 0.081), indicating that PW nurses’ stress was a significant determinant of depressive symptoms. The standardized total effect of stress affected depressive symptoms through the mediating roles of quality of sleep and occupational burnout (z = 3.125, *p* < 0.001) (Table 1). In the final model, stress, quality of sleep, and occupational burnout explained 46.0% of variance in PW nurses’ depressive symptoms (Figure 2).

## 4. Discussion

For PW nurses, perceived stress was not directly related to depressive symptoms, but it affected quality of sleep and occupational burnout directly. In the model of parallel multiple mediators for depressive symptoms, quality of sleep and occupational burnout played mediating roles, and these two mediators strengthened the effect of stress on depressive symptoms in this model. As a result, stress did not affect depressive symptoms directly, but it affected depressive symptoms through the mediating effects of quality of sleep and occupational burnout. The results contributed to understanding the influential mechanism of stress on depressive symptoms.

In our study, the effect of perceived stress on depressive symptoms was through the indirect path of quality of sleep. Our result was similar to a previous study finding that poor quality of sleep had indirect effect on depressive symptoms [13]. Stress perception is a common issue among nurses, especially among PW nurses who experience a wide range of stressful events related to the care of PW patients with violence, recurrent relapse, and poor prognoses [27]. Previous studies have revealed that stress is a leading cause of poor quality of sleep [28] and depressive symptoms [2], and our results were similar to these. Some studies have suggested that as a result of continuous shift work, the daily schedules of most nurses change constantly, which has led to sleep disturbance problems [2,28]. One study found that the risk of nurses developing major depression was 1.5 times higher than that of other professionals [11].

There are some possible reasons for the completed mediation effect of quality of sleep. Besides the issues related to shift work, PW nurses work in a highly stressful environment for long hours at a time, and even when they are off duty, some nurses still ponder work issues, worry about the condition of a patient the next day, or think about negative incidents that had occurred during the day, such as attacks or threats from a patient or involvement in a care-related issue. This form of thinking not only causes muscle tension and anxiety, but also hinders the nurses’ ability to fully relax, thereby affecting their quality of sleep. Often nurses do not get enough sleep because they have to provide care for children and elderly family members at home after work. According to a survey conducted by the Ministry of Health and Welfare, currently more than 80% of home caregivers in Taiwan are women [29]. Taiwanese women work both in and out of the home, because it is considered “expected” and “granted” for women to look after families in Taiwan. Interventions targeted at improving quality of sleep and reducing sleep disorder are needed to improve depressive symptoms among PW nurses.

In addition, the effect of perceived stress on depressive symptoms was also through the indirect path of occupational burnout. This result was supported by previous studies that found occupational burnout had indirect effect on depressive symptoms [30]. Previous studies have demonstrated that when exposed to a stressful work environment, nurses will likely develop occupational burnout, which can result in chronic fatigue and depressive symptoms [30,31]. PW nurses devote themselves to taking care of patients with mental disorders, and this takes them much time, extreme patience, and great compassion; however, in the long run, such situations might make nurses feel tense, distressed and irritable, and prone to develop occupational burnout symptoms [14]. Compared with nurses in other departments, PW nurses are far more likely to suffer from depressive symptoms [32].

Two reasons were postulated for the completed mediation effect of occupational burnout. Firstly, in this study, PW nurses’ client-related burnout was exposed to a highly stressful work environment, as they provide care to patients with complex and challenging conditions who might even demonstrate a tendency toward violence or suicide. In addition, these patients are often admitted repeatedly. Therefore, PW nurses usually lack a sense of achievement during the care process.

Secondly, most Taiwanese have different viewpoints and reactions from people in western countries towards having a family member with mental disorders. For example, some superstitious family members of patients with mental disorders consider that the cause of mental disorder is “the person being possessed by a demon” (the patient has been possessed by a ghost). They needed to be treated by “charm water (符水)” or the therapy of “recovering lost souls (收驚)” by specific ritual behaviors, etc. [33]. Some families of the patients might even exhibit verbal or physical violence towards the PW nurses, if they are not pleased by treatment inconsistent with their expectations.

However, the PW care model is primarily based on the work of a team composed of physicians, nurses, psychologists, social workers, and occupational therapists. The lack of positive team collaboration may also add to the stress and burden of PW nurses. This notion was supported by a previous study [14] that indicated occupational burnout and insufficient resources might worsen PW nurses’ depressive symptoms. The findings of this study pointed out the influential mechanism of developing depressive symptoms among PW nurses, and it can be a reference material for future policy by hospital managers. In order to improve quality of sleep, and alleviate burnout and depressive symptoms among PW nurses, the following interventions are proposed for future development: Firstly, hospital managers should reduce the number of shifts when arranging rosters, thereby allowing PW nurses to obtain adequate sleep and rest. Secondly, the PW care mode is mainly based on teamwork, so peer assistance and understanding among team members should be promoted, thereby alleviating their stress of patient care. Thirdly, hospital managers need to focus on how to support PW nurses, provide opportunities to build knowledge, skills, and a safer working environment to reduce violent incidents in their caring for patients.

Fourthly, as sufficient manpower is required in the PW wards to prevent these violent incidents, the nurse–patient ratio should be raised by hospital managers to 1:5 or above to reduce work stress and occupational burnout. Finally, to facilitate the establishment of a good relationship between nurses and patients, arrangements should be made whereby PW nurses can look after the same group of patients during their shifts, allowing them to better understand the patients’ conditions and prevent accidents.

## 5. Conclusions

This study investigated the relationships between depressive symptoms and sleep quality and occupational burnout among PW nurses. The results contribute to our understanding of the influential mechanism of stress on their depressive symptoms. The parallel multiple mediator models in this study can potentially contribute to the prevention and intervention of burnout in PW nurses. The completed mediation effects of quality of sleep and occupational burnout were confirmed. Perceived stress was not directly related to depressive symptoms, but it affected quality of sleep, occupational burnout, and depressive symptoms directly. Stress affected PW nurses’ depressive symptoms simultaneously through quality of sleep and occupational burnout. Quality of sleep and occupational burnout were major determinants of depressive symptoms.

## Figures and Tables

**Figure 1 ijerph-18-07327-f001:**
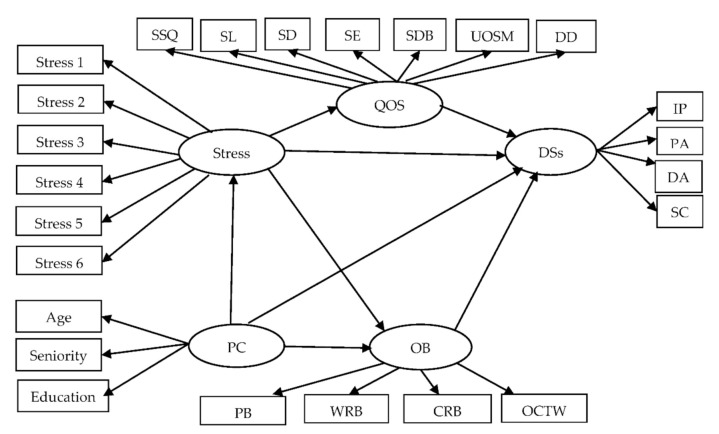
Conceptual framework of hypothesized model. Note. Personal characteristics = PC; quality of sleep = QOS; depressive symptoms = DSs; occupational burnout = OB; subjective sleep quality = SSQ; Sleep latency = SL; sleep duration = SD; Sleep efficiency = SE; sleep disturbance = SDB; use of sleeping medication = UOSM; Daytime dysfunction = DD; interpersonal problems = IP; positive affect = PA; depressed affect = DA; somatic complaints = SC; personal burnout = PB; work-related burnout = WRB; client-related burnout = CRB; over-commitment to work = OCTW; 
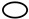
 = latent variable; 
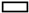
 = measured variable; → = unidirectional path.

**Figure 2 ijerph-18-07327-f002:**
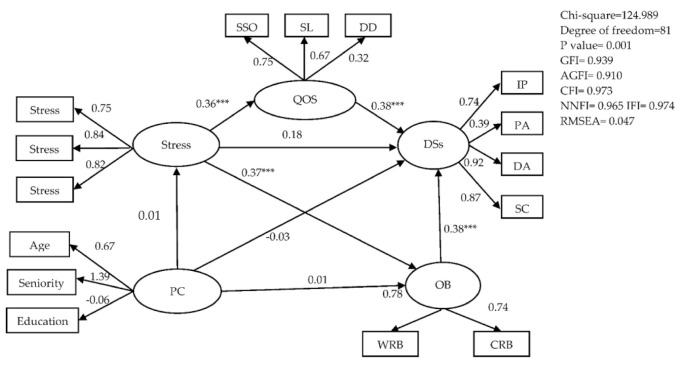
Conceptual framework of hypothesized model. Note. Personal characteristics = PC; quality of sleep = QOS; Depressive symptoms = DSs; occupational burnout = OB; subjective sleep quality = SSQ; sleep latency = SL; sleep efficiency = SE; daytime dysfunction = DD; interpersonal problems = IP; positive affect = PA; depressed affect = DA; somatic complaints = SC; personal burnout: PB; work-related burnout = WRB; client-related burnout = CRB; over-commitment to work = OCTW; 
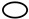
 = latent variable; 
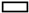
 = measured variable; → = unidirectional path; *** = *p* < 0.001.

**Table 1 ijerph-18-07327-t001:** Parallel multiple mediator model for stress on depressive symptom.

Relationship	Point Estimate	Product of Coefficients		Percentile		Bootstrapping		
				95% CI		BC 95% CI		
		SE	Z	Lower	Upper	Lower	Upper	*p*
	Indirect Effects							
S to QOS to D	0. 449	0.192	2.931	0.145	0.881	0.167	0.931	<0.001 ***
S to OB to D	0.448	0.189	3.285	0.141	0.876	0.179	0.963	<0.001 ***
	Direct Effects							
S to D direct	0.567	0.323	1.755	−0.081	1.213	−0.087	1.210	0.081
	Total Effects							
S to D total	0.897	0.287	3.125	0.412	1.572	0.453	1.633	<0.001 ***

Note. Stress = S; depressive symptom = DS; quality of sleep = QOS; occupational burnout = OB; SE = standard error; 1000 bootstrap samples. *** *p* < 0.001.

## Data Availability

Not applicable.

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
