# Peer review of "Relations between Stress and Depressive Symptoms in Psychiatric Nurses: The Mediating Effects of Sleep Quality and Occupational Burnout"

_ijerph, 2021, doi:10.3390/ijerph18147327_

Round 1

Reviewer 1 Report

The study is based on advanced statistical methods that I am not familiar with and unable to give a meaningful review on. I just have some minor remarks of editorial type to provide:

Several of the references are given in the text and identified by year of publication rather than number in the reference list, some of them even not included in the reference list. All reference should be included in the reference list and referred to by reference list number in the text.  This applies to references in lines 40, 50, 100, 116, 118, 129, 139, 148,161, 168, 268, 294.

In Figure 1, top row, SSO should be SSQ, and UOS should be UOSM to correspond with the figure text.

Line 105: ...principal investigator (PI)...      Who is PI? Please indicate with author list on the front page.

Lines 154-159: The total..........present study.   isn't this results from analysis of the data which should rather be presented in the results chapter?

Line 175: 'Statistics-wise'   If the meaning is mean values, say so.

Author Response

The study is based on advanced statistical methods that I am not familiar with and unable to give a meaningful review on. I just have some minor remarks of editorial type to provide:

Q1: Several of the references are given in the text and identified by year of publication rather than number in the reference list, some of them even not included in the reference list. All reference should be included in the reference list and referred to by reference list number in the text.  This applies to references in lines 40, 50, 100, 116, 118, 129, 139, 148,161, 168, 268, 294.

A1: Thank you for your recommendation, we have revised our reference list, and corresponding number were given in the text.

Q2: In Figure 1, top row, SSO should be SSQ, and UOS should be UOSM to correspond with the figure text.

A2: Thank you for your recommendation, we have revised SSO to SSQ and UOS to UOSM (Figure 1).

Q3: Line 105: ...principal investigator (PI)...      Who is PI? Please indicate with author list on the front page.

A3: The first author was the principal investigator, we indicated it in the part of “Data Collection and Ethical Considerations” (Page 3).

Q4: Lines 154-159: The total..........present study.   isn't this results from analysis of the data which should rather be presented in the results chapter?

A4: Thank you for your recommendation, we have moved the lines 154-159 to the results chapter (Page 4).

Q5: Line 175: 'Statistics-wise'   If the meaning is mean values, say so.

A5: Thank you for your recommendation, they are mean values. For avoiding misunderstanding, we have deleted the word 'Statistics-wise' (Page 4).

Reviewer 2 Report

Dear authors,

Thanks for providing us with interesting and well-conducted research. Nurses are in the frontline, not only in the COVID era, experiencing major depressive symptoms which constrains their quality of life and wellbeing.

I believe that your paper is a strong contribution to the field, and the research design and methods are well documented. I suggested improving the introduction section with more references since in the current one there are a small number of them. Also, in the results section, please improve the presentation of the figures.

Good work!

Author Response

Thanks for providing us with interesting and well-conducted research. Nurses are in the frontline, not only in the COVID era, experiencing major depressive symptoms which constrains their quality of life and wellbeing.

I believe that your paper is a strong contribution to the field, and the research design and methods are well documented.

Q1: I suggested improving the introduction section with more references since in the current one there are a small number of them.

A1: Thank you for your recommendation, we have added some reference in the introduction section (Page 2).

Q2: Also, in the results section, please improve the presentation of the figures.

A2: Thank you for your recommendation, we have revised the figures (Figure1 and Figure 2).

Reviewer 3 Report

This article provided a quantitative and insightful investigation of the sources and origins that affected the depressive symptoms in psychiatric ward nurses of south Taiwan. Whilst the results were not particularly surprising (sleep quality and occupational burnout being the primary factors), the methodology and statistics have been strictly employed during the investigation. More importantly, the investigation properly pointed out that perceived stress served as mediators for depressive symptoms. Furthermore, there were extended sections discussing the local cultural factors that impacted the occupational burnout level, which was both meaningful and crucial to the readers from broader cultural backgrounds. Finally, the manuscript was well-written with elegant language.

Author Response

This article provided a quantitative and insightful investigation of the sources and origins that affected the depressive symptoms in psychiatric ward nurses of south Taiwan. Whilst the results were not particularly surprising (sleep quality and occupational burnout being the primary factors), the methodology and statistics have been strictly employed during the investigation. More importantly, the investigation properly pointed out that perceived stress served as mediators for depressive symptoms. Furthermore, there were extended sections discussing the local cultural factors that impacted the occupational burnout level, which was both meaningful and crucial to the readers from broader cultural backgrounds. Finally, the manuscript was well-written with elegant language.

A: Thank you for your comments!

Round 2

Reviewer 1 Report

As I said in my first report, the study is heavily based on statistical methods that I am not familiar with and the presentation and discussion of which I am unable to evaluate and unable to tell whether the choice of method,, the presentation of method or their conclusions are adequate.  The authors have adequately made the minor editorial changes that I suggested in my first report, but there is still a need for language check, mostly for occasional grammatic  errors.  Because of the above, I am reluctant to give a recommendation for or against acceptance, and when I mark the 'accept after minor...' button , that is only because your system require an answer to accept submission of the report.

I repeat my previous recommendation that the article should be reviewed by someone well trained in advanced statistics

Author Response

Q1: As I said in my first report, the study is heavily based on statistical methods that I am not familiar with and the presentation and discussion of which I am unable to evaluate and unable to tell whether the choice of method, the presentation of method or their conclusions are adequate.  The authors have adequately made the minor editorial changes that I suggested in my first report, but there is still a need for language check, mostly for occasional grammatic errors.  Because of the above, I am reluctant to give a recommendation for or against acceptance, and when I mark the 'accept after minor...' button, that is only because your system require an answer to accept submission of the report.

A1: Thank you for your recommendation, my manuscript has been already edited by an editing company.

Q2: I repeat my previous recommendation that the article should be reviewed by someone well trained in advanced statistics

A2: Thank you for your recommendation, I hope that this problem can be solved by the editor and I will follow the editor's opinion.
